# Adaptive volumetric light and atmospheric scattering

**Tan shihan***, **Zhang jianwei, Lin yi, Liu hong, Yang menglong, Ge wenyi**

The National Key Laboratory of Fundamental Science on Synthetic Vision, Sichuan University, Chengdu, Sichuan province, China

* 530916232@qq.com

**Data Availability Statement:** All relevant data are within the manuscript and its Supporting information files.

**Funding:** The author(s) received no specific funding for this work.

## Abstract

An adaptive sampling-based atmospheric scattering and volumetric light framework for flight simulator (FS) is proposed to enhance the immersion and realism in real-time. The framework comprises epipolar sampling (ES), visible factor culling (VFC), interactive participating media density estimating (IPMDE). The main process of proposed architecture is as follows: the scene is divided into two levels according to the distance from the camera. In the high-level pipeline, the layer close to the camera, more samples, and smaller sampling step size is used to improve image quality. Further, the IPMDE method is designed to enhance realism by achieving interactivity with the participating media and multiple light sources. Further optimization is performed by the lookup table and 3D volumetric textures, by which we can compute the density of participating media and the scattering coefficient in parallel. In the low-level pipeline, when the samples are far away from the camera, its influence on the final output is also reduced, which results in fewer samples and a bigger sampling step size. The improved ES method further reduces the number of samples involved in ray marching using the features of epipolar geometry. It then generates global light effects and shadows of distant terrain. The VFC method uses an acceleration structure to quickly find the lit segments which eliminate samples blocked by obstacles. The experimental results demonstrate our architecture achieves a better sense of reality in real-time and is very suitable for FS.

## Introduction

This chapter presents adaptive volumetric light and atmospheric scattering, a technique developed for FS. In this paper, we propose a new adaptive volumetric light model and an improved ES algorithm by using a simpler and faster 1D min-max mipmaps, which enables us to incorporate volumetric light into integral, while maintaining real-time performance. We observe that by applying LOD (level of detail) and adaptive sampling, the fixed sampling interval is not indispensable when the camera frustum is zoomed out, which allows us to use fewer samples without sacrificing the image quality. The framework comprises epipolar sampling (ES), visible factor culling (VFC), interactive participating media density estimating (IPMDE). Among them, adaptive sampling framework and IPMDE are proposed in this paper. The ES and VFC are inherited from other researchers [1, 2] in which we have made targeted modifications to

**Competing interests:** The authors have declared that no competing interests exist.

adapt to FS or other outdoor open space applications. The workflow of proposed architecture is as follows: In the low-level pipeline (far away from the camera), the improved ES smartly marches along the epipolar lines instead of performing expensive pixel-by-pixel traversal. Then, the VFC is used to eliminate samples blocked by terrain or other obstacles that further promote real-time performance. At the core of the VFC algorithm is the use of an acceleration structure (a 1D min-max binary tree), which allows us to quickly find the lit segments for all pixels in an epipolar slice in parallel. Finally, in the high-level pipeline, the IPMDE algorithm is proposed to support the effect of multiple light sources and volumetric fog near the camera, which achieves a better immersion and interaction by using series optimized methods and reasonable sampling strategy.

The proposed framework renders atmospheric scattering and volumetric light, which is essential for producing compelling virtual scenes. However, simulating all the scattering events is prohibitively expensive, especially for real-time applications. The most crucial advantage of proposed method is the introduction of adaptive sampling for volumetric light, which assigns samples involved in ray marching according to the distance from the camera and weight of the final output effect. Moreover, a series of subsequent optimizations can be performed based on this framework. In short, our original contributions can be described as follows:

a) Our technical contribution is to eliminate avoidable samples, replacing it with a larger sampling intervals according to the distance from the camera, which allows all samples effective for the final result. After adaptive sampling, a smooth transition between different pipelines is applied to accumulate the color and transparency.

b) We observe that by applying epipolar geometry to the shadow map, each camera ray only travels through a single row of the shadow map, which allows us to find the visible factor by considering only 1D depth fields. Then, we convert all the visibility tests within a 1D depth map without rectification, which allows us to avoid the singular values close to the epipolar line.

c) A participating media density estimating algorithm is proposed to enhance distance perceive by introducing variable density of participating media and multiple light sources. To achieve good performance, we use the 3D volumetric texture for hardware acceleration, which allows us to compute participating media density and scattering coefficient in parallel.

## Related works

**The atmospheric scattering model.**　An atmospheric stratification model was proposed [3] and summarized [4] in which the atmosphere was divided into parallel planes assuming the same density in each layer. Kaneda K et al. [5] conceived a new algorithm [6, 7] regarding the atmosphere as a spherical shell where the density of air particles was exponentially decreasing along with the altitude. The model of Rayleigh scattering was studied [8] using an empirical function. To accurately simulated the atmospheric scattering under different weather conditions, a more complete model was developed [9]. Qualitative and quantitative evaluation was given [10] for a clear sky. Later contributions were made by Zuliang A et al. for introducing the sky illumination model by multiple scattering [11]. The papers above were dedicated to solving the scattering model and integral. However, these methods did not take shadowing into account, which was often required for realism and was the effect on which we concentrated.

**Pre-calculation and lookup table.**　Nishita et al. described how to optimize the performance of optical depth integral using lookup tables [8]. The 3D lookup table was then

proposed [12] for volumetric clouds by introducing depth value as the third dimension. Further, Neyret et al. designed a 4D parameter table [13], which simulated multiple scattering and reflection at the same time. Later contributions were made by Klehm et al. using the prefiltered method.

**Epipolar sampling.**   Epipolar sampling method was proposed [1] which sampled sparsely along epipolar lines and interpolated between samples. The scattered radiance at all other points was interpolated along these lines, causing temporally-varying artifacts.

**Volumetric light.**   The volumetric light was the same as obtaining 2D medical images from 3D magnetic resonance(MRI) or computed tomography (CT) data. The primary previous studies on volumetric light were found in References [14–19]. Later improvement was proposed by volumetric shadow method, which can be combined with epipolar sampling for further acceleration. [20–22]. Single scattering with inhomogeneous media was studied [23] by shadow map-based polygon mesh which took into account the interaction with inhomogeneous media. A later contributions to this approach were made [24] to generate volumetric fog effect. Then the Monte Carlo algorithm and neural networks were also used [25] to achieve volumetric cloud and light. Still, these methods were expensive and we did not observe a significantly realistic improvement.

**1D min-max binary tree.**   Previous studies of data structure emphasize potential advantages to visibility test, such as 1D min-max mipmaps [2] by Chen et al. However, this implementation does not support all directions from camera ray and to compare with their method, we used a more efficient approach without epipolar rectification. Tevs et al. [26] also reap huge fruits from 1D min-max mipmaps. However, it mainly focused on large-scale height fields of terrain system.

**Voxelized shadow volumes.**   According to the similar epipolar space sampling, voxelized shadow volumes [27] played an important role in the evolution of performance. Later contributions were made by the same team using the improved VSV [28], by which the approach could be scaled to many lights. However, this implementation aliased near singularity and needed to pay more attention for robustness. To compare with their method, we used a shared component of GPU, which allowed us to reap huge fruits from sharing of intermediate results.

**Image-based methods.**   The image-based methods [29, 30] used fewer sampling points to obtain the same quality output by anti-aliasing and interpolation. More contributions to this approach were proposed to reduce samples and deformation marching along the ray [31]. Further improvement for soft filtering shadows was studied [32–35] using mipmaps. Other atypical methods were proposed [36–38] including radiance transfer, compressed light fields and ray-box. However, these methods cannot handle object boundaries and shadows very well.

## Overview

Atmospheric scattering effects and volumetric light are paramount to create the realism of the outdoor scenes. It also generates the a clean sky as well as the optical phenomenon at different time of day. Crepuscular rays or god rays are also performed by the similar optical model in the presence of obstacles. Facing massive computation and complexity by the nested integral, series of simplified models are studied, such as analytical exponential fog, screen space-based solutions, methods based on artistic configuration, and particle systems. However, only simplified light models are utilized in real-time rendering, which has a lot of disadvantages. Computer graphics researchers try to use a more precise model to generate those effects by which can not only significantly enhance the realism of the scene, but also establish a visual distance perception between objects and light. For real-time rendering applications such as FS, it is

usually necessary to simplify and approximate the model. As introduced in the related work in the previous section, the rendering of atmospheric scattering and volumetric light includes image-based methods, particle and billboard methods driven by artistic effects, and modern ray marching based solutions. Various methods have their advantages and disadvantages. However, we find that ray marching-based approaches are strictly in accordance with the complete physical model and can be performed by modern GPU in parallel. With hardware developments, more programmable pipelines are available, which prompts researchers to develop intelligent approaches for solving the existing problems. Consequently, an improved ray marching-based approach, which is used to approximate optical model, is an indispensable part and a strong support of the proposed architecture.

Studies of the ES approaches suggest that by applying the epipolar geometry, they perform sampling only along the epipolar lines instead of per-pixel calculation, while maintaining the same visual quality. The remaining pixels can be obtained by interpolation. However, this implementation does not support multiple lights and to compare with their method, the proposed IPMDE approach is applied to interact with variable density of participating media by many light sources.

The 1D min-max mipmaps method [2] can significantly improve the efficiency of visibility test and avoid the occluded sampling points from participating in expensive ray marching. Comparatively, the 1D min-max mipmap in this paper is significantly improved by using adaptive sampling. Moreover, to avoid rectification that leads to paying careful attention to the area near the epipolar line, we implement it with a general and efficient way.

The work proposed [27] by Chris Wyman, used a very similar approach by epipolar space sampling, which aligns the samples of ray marching in memory. Then, many lights are supported by their subsequent works [28]. According to the VSV approaches, epipolar space sampling also plays an important role in the evolution of performance, which is described in the previous section. However, we find that some samples near the singular point are not strictly in accordance with the regulations. In this paper, we achieve a significant improvement by using a universal component shared by many stages, which allows us to prevent from aliasing near to the singular point.

Studies of interaction between air medium and volumetric light plays a paramount role in this paper. However, we find that the existed approaches are not strictly in accordance with the regulations and only contain uniform density of participating media. For our proposal of IPMDE, by introducing variable density model of media instead of volumetric Perlin noise [39] and Gaussian noise [40], we enhance the immersion and realism, which allows us to construct a visual distance perception. Moreover, pseudo spheroids and the Gaussian blob model are introduced to avoid repeated calculations when air particles overlap. By applying pseudo-spheroids with a radius, we can easily count its contribution to the shadow.

At the beginning of our study, we tried to use existing volumetric light and ray marching methods for FS. However, as mentioned above, existing methods have their inherent limitations. We summarized as follows:

a) Based on existing LOD approach, the rendering of terrain and 3D models can easily manage the scene complexity, which reduces the number of triangular strips far from the camera when the camera is zoomed out. Unfortunately, the existing approaches do not take into account the LOD and adaptive sampling for volumetric light, which leads to maintaining real-time performance, remains challenging.

b) Note that for complex scenes, the existing ES can only support sun as the single light source and leads to incorrect interaction between multiple lights and objects.

 c) According to the 1D min-max mipmaps, epipolar rectification plays an important role in the VFC method. However, it is sophisticated and lies upon a singular value decomposition of the scattering term, which leads to paying careful attention to the area near the epipole line.

 d) From the perspective, the density of participating media must change according to the external environment and user control. However, the existing approaches do not support variable density model, which leads to monotonous and unreal scene.

To address the above problems, we propose the adaptive sampling-based framework. The ultimate goal of the proposed system is to allow us to use fewer samples along epipolar lines without sacrificing image quality. At the core of our algorithm is the use of improved ES and 1D min-max mipmaps, which allows us to quickly find the lit segments within each epipolar slice in parallel. Our technical contribution is to eliminate the epipolar rectification used for integration, replacing it with a quick min-max mipmaps, allows all rays to be processed in parallel.

## Materials and methods

The proposed framework comprises: epipolar sampling (ES), visible factor culling (VFC) and interactive participating media density estimation (IPMDE). In the first place, we start from the screen space and accumulate the light intensity reaching the screen pixel. To avoid performing all the screen pixels, we utilize the characteristics of epipolar geometry that the intensity of scattered light changes uniformly along the epipolar line. There is one more point that we need to project a ray from the camera through each epipolar sample. Then, We convert each view ray to shadow map space and perform a visibility test by the proposed VCF, which prevents the occluded ray segments from participating in the calculation. The last but not the least, the proposed IPMDE method realizes more realistic and detailed interactive features by supporting variable participating media density.

As shown in Fig 1, the proposed architecture uses the following per-frame procedure:

1. According to the distance from the camera, we divide the camera perspective space into two levels.

2. In the low-level pipeline, fewer samples and bigger sampling step size are used.

3. Marches along the epipolar lines on the screen instead of performing on every screen pixel by the ES method.

4. Eliminate samples blocked by terrain or other obstacles by an acceleration structure (a 1D min-max binary tree).

5. In the high-level pipeline, more samples and smaller sampling step sizes are used for ray marching which provides more details of volumetric light.

6. Change the density of the participating media by the proposed model in the IPMDE subsystem.

7. The light from multiple sources interacts with variable density of participating media.

8. Speed up the calculation of ray marching by 3D volumetric texture and other optimizations.

9. Perform ray marching on each view ray, and accumulate the transparency of each sample. When the accumulated transparency reaches 1, the traversal is terminated.

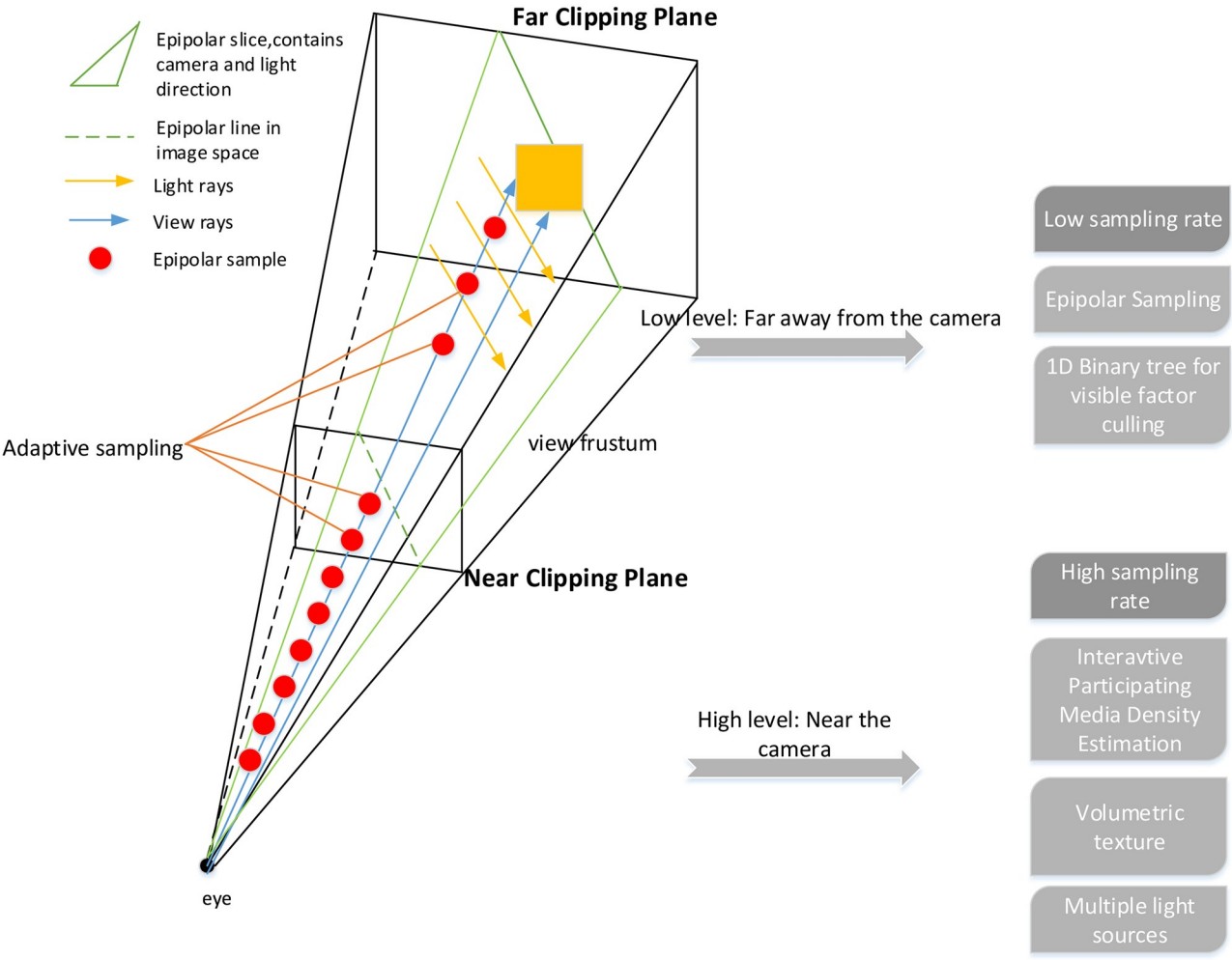

**Fig 1. Adaptive volumetric light architecture.**

Compared with the traditional method, according to the distance from the camera, we reasonably designed the sampling step size and its distribution, which further formed the proposed adaptive sampling framework as described in Algorithm 1.

**Algorithm 1** Adaptive sampling of proposed framwork

```
// TN:total number of samples; RatioL1: ratio of samples in L1 level
T1 = TN * RatioL1
// T1: number of samples in L1; T2: number of samples in L2
T2 = TN - TN * RatioL1
// NCP:near clip plane; FCP: far clip plane; StepL1Base: fixed step of
samples in L1
StepL1Base = (NCP - 0)/T1
StepL2Base = (FCP - NCP)/T2
for i = 0 to T2 - 1 do
  //StepL2[i]: every sample in L2 using adaptive step based on
StepL2Base
  StepL2[i] = StepL2Base * (i - T1)/T2 * 2/(T2 + 1)
end for
```

## Epipolar sampling

In practice, there are two main types of problems that are associated with existing ES method, and they constitute a bottleneck in sample distribution and need to be addressed with the proposed approach. The first and foremost, by applying variable density of participating media, we construct clear sense of distance. There is one more point that we support multiple light sources instead of a single light source by sun, which allows volumetric light to interact with objects of various scenes correctly.

In this section, we describe the proposed sampling strategy that determines where the inscattering term is computed, and where it is interpolated from nearby samples as summarized in the following steps:

a) A configuration of epipolar lines is defined, then sampling refinement of initial equidistant sampling along these lines is applied to capture depth discontinuities.

b) Perform ray marching along the view ray and accumulate the color and transparency.

c) Interpolation is performed along and between the epipolar line segments.

d) The color and transparency of multiple light sources are accumulated into a 2D texture which is mapped to the screen space.

The generation of the epipolar lines and samples is the most important process shown in Fig 2. When sun position is within the screen, we place the samples and initial point for each line where it should be (Fig 3 Bottom left). When the sun is off the screen, we place the original point at the intersection of the epipolar line and the boundary (Fig 3 Top left). The intersections of these epipolar slices (planes) with the image plane are epipolar lines, along which streaks of lighting appear to emanate. We place the epipolar sampling points evenly along each epipolar line. The results of ray marching only depend on the epipolar samples within the same slice, which implies that we can process the slices in parallel.

The sample refinement is performed by searching for depth discontinuities using the depth buffer of the scene. After placing and refining the sampling points, the interpolation is performed to reconstruct the in-scattering for each pixel of the final image. Finally, the results are used together with the 2D texture from the IPMDE method to accumulate color and transparency.

Our technical contribution is to eliminate the single light sources used for integration, replacing it with a multiple light sources, which allows all rays to be processed in parallel according to the scene, as shown in Algorithm 2.

**Algorithm 2** Ray marching of the improved ES

```
// Calculate the pos of current samples in the world coordinate
WorldPos.xyz = CalculatePos(ThreadID.xyz);
// Calculate the horizontal density at the same height using IPMDE
HorizontalDensity = CalculateHorizontalDensity(WorldPos);
// Calculate the scattering coefficient,
RayDir.xyz = (Objext.xyz – Camera.xyz)/NStep;
ds = ||RayDir.xyz||;
// Initial vertical density, rlgh and mie scattering
VerticalDensity = 0;
for i = 0 to NStep do
  CurrentPos = Object.xyz+ RayDir.xyz*i;
  h = abs(CurrentPos – EarthCenter) – EarthRadius;
  //Calculate the final density by combining height and horizontal
density changes
  VerticalDensity = e^{−h/H};
```

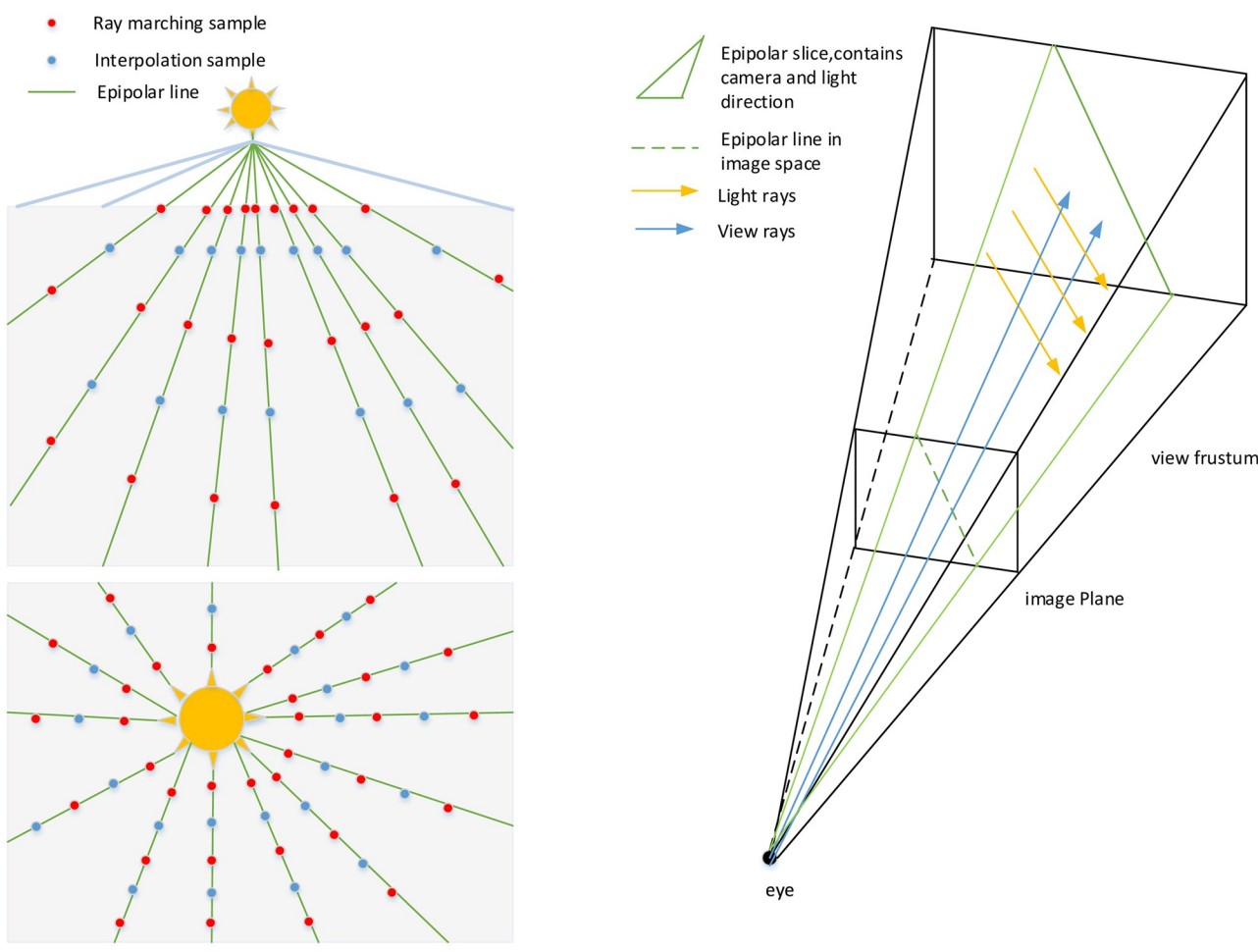

**Fig 2. Ray march sampling and epipolar slice.**

```
//Generate lookup table for density from the surface to the top of
the atmosphere
```
$\cos\varphi = NEarth.xyz - SunDir.xyz$;
$VerticalDensity_{A \to P} = D[h, \cos\varphi]$;
```
// Accumulate the participating media density
```
$VerticalDensity_{P \to C} += VerticalDensity^* ds$;
$VerticalDensity_{A \to P \to C} = VerticalDensity_{A \to P} + VerticalDensity_{P \to C}$;
$if(CurrentPos.z < Threshold)$
```
// Accumulate horizontal density changes near ground
```
$FinalDensity = VerticalDensity_{A \to P \to C} + HorizontalDensity$;
```
// Calculate the optical depth
```
$OpticalDepth = FinalDensity * \beta^e_{RM}.xyz$;
$Attenuation = e^{-(T_{R.xyz} + T_{M.xyz})}$;
$DiffScattering = FianleDensity * \beta^s_{RM}.rgb + Attenuation.rgb * ds$;
$FinalScattering += DiffScattering^* Visibility$;
**end for**
```
// Calculate the contribution of multiple light sources
```
$SumLight = GetSunLight(WorldPos)^* PhaseFunction(Raydir, SunDir)$;
**for** $j = 0$ to $NumLights$ **do**

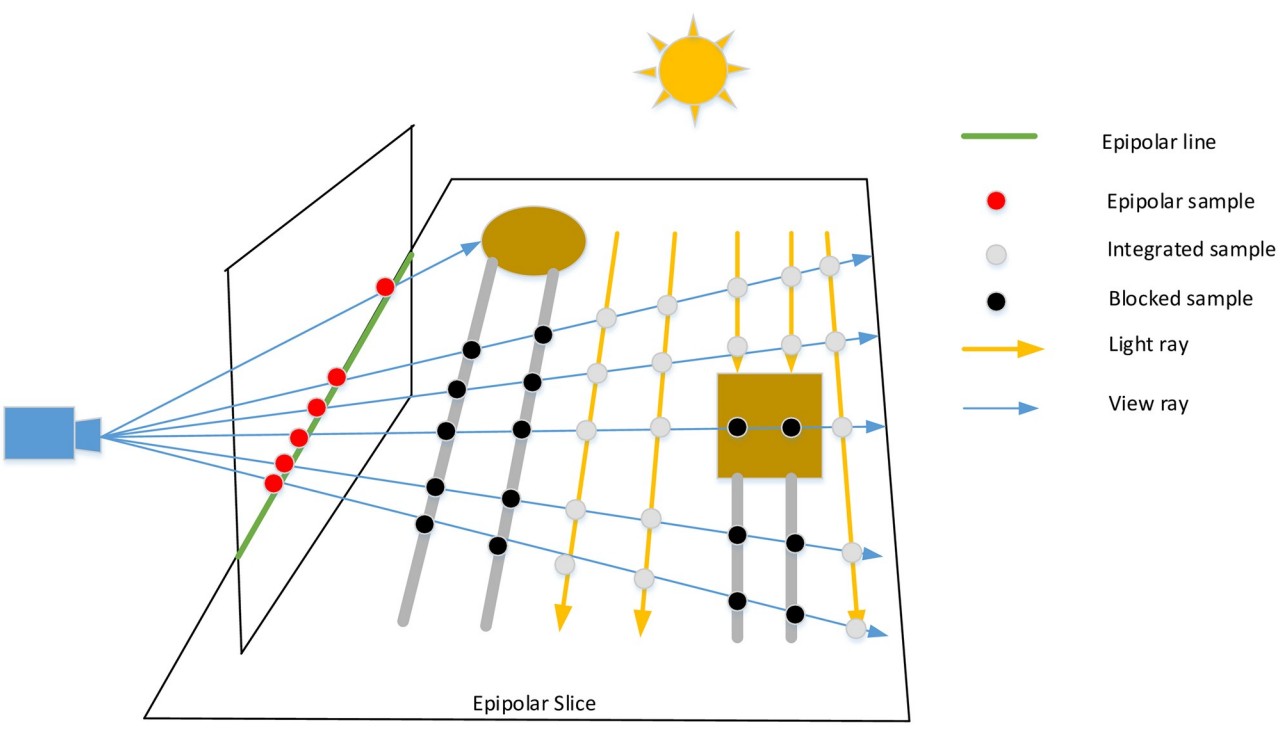

**Fig 3. Rays from camera through the samples of the epipolar line.**

```
    SumLight+ = GetLocalLight(j, WorldPos)*PhaseFunction(Raydir,
LightDir);
end for
FinalOutput = vec4(SumLight*FinalScatteirng.rgb, Attenuation);
```

## Visible factor culling

At this point, we finish the selection of sampling points on each eippolar line and compute the scattering integral, while taking shadow into account. According to VFC, 1D min-max mip-maps introduced by Chen et al. plays an important role to quickly find lit segments for all view rays. However, it is rather sophisticated and depends on a singular term disintegration of the scattering term, which leads to paying more attention to the region close to the epipolar line. Our main contribution is to eliminate the epipolar retification, replacing it with a static and simple manner, which allows for massive parallelism. The pipeline of VFC can be described as follows: In the first place, we project the view ray to the shadow map space where camera rays cat throught samples on epipolar lines. There is one more point that the view rays in the shadow map are defined by the initial point and direction. The last but not the least, the min-max binary tree is constructed and traversed.

We traverse every sample by casting a ray, transforming its begin and exit positions to the camera space, then sampling and calculating internal scattering within shadow-map coordinate system. We apply a 2D lookup table to compare the depth between the samples and the occlusions. Only the lit samples (the depth of the current sample less than that of the obstacle) is accumulated along the ray. In contrast, shadowed samples are prevented from participating in ray marching, as shown in Fig 3.

We take samples in shadow-map coordinate system along the epipolar line resulting in a 1D depth map. Every camera ray within the epipolar slice is converted into this space. To

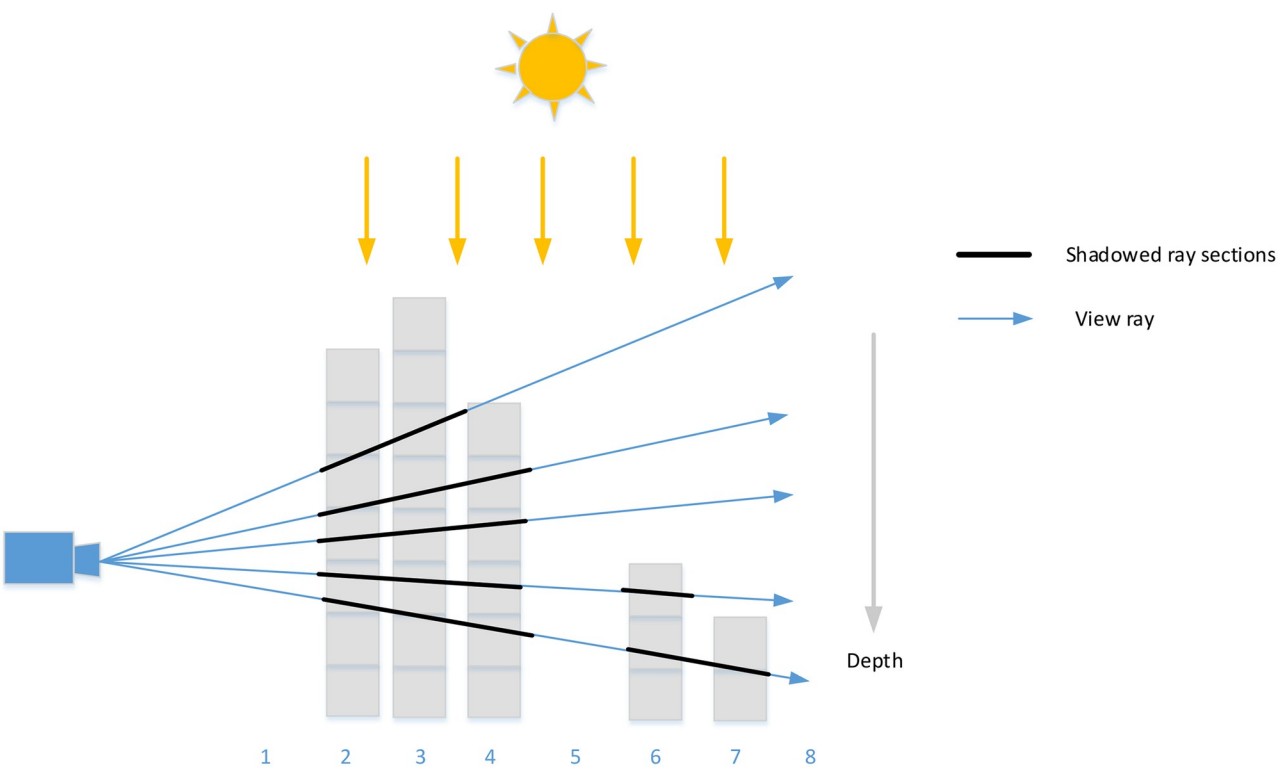

**Fig 4. 1D depth map for detecting lit and showed ray sections.**

perform a visible factor culling, we necessarily need to detect if the present samples along the camera ray are under the depth map or above it. As shown in Fig 4, the blue lines are view rays, while the black ones are segments in shadow, which means the current sample is above the depth map (Depth and height are inversely proportional).

To construct the acceleration structure, the intersection between the epipolar slice and the shadow map needs to be defined. View rays within the epipolar-slice are projected to the epipolar line by using whichever two samples along the ray and converting them to the shadow-map coordinate system. The camera position is regarded as the original point. And its projected location $O_{uv}$ represents the zeroth sample at the one-dimensional depth map. The exit point on the ray is cast through the end sample of the slice, converted into the uv space. The direction $D_{uv}$ is calculated from the start point to the final point. Then we normalize $D_{uv}$ to fit subsequent calculations. If $O_{uv}$ falls outside the boundary, we shift the initial point to the first intersection with the boundary (See Fig 5).

At this point, we are conscious of the location $O_{uv}$ as the zeroth point in the one-dimensional depth map and its direction $D_{uv}$. We are able to deduce the position of ith samples as $O_{uv} + i * D_{uv}$. For every single epipolar slice, we store its depth as depth[i]. By comparing the value of $O_{uv} + i * D_{uv}$ and depth[i], we can determine whether the current samples are lit or shadowed.

The upper layer of the binary tree is constructed by calculating the min-max value of every 2ith and (2i+1)th sampling points. Then, the whole tree is constructed by spreading these min/max nodes upwards (See Fig 6a).

At this point, we have constructed the acceleration structure and need to perform the visibility test. As shown in Fig 6, for the current ray sections, we extract its maximum and

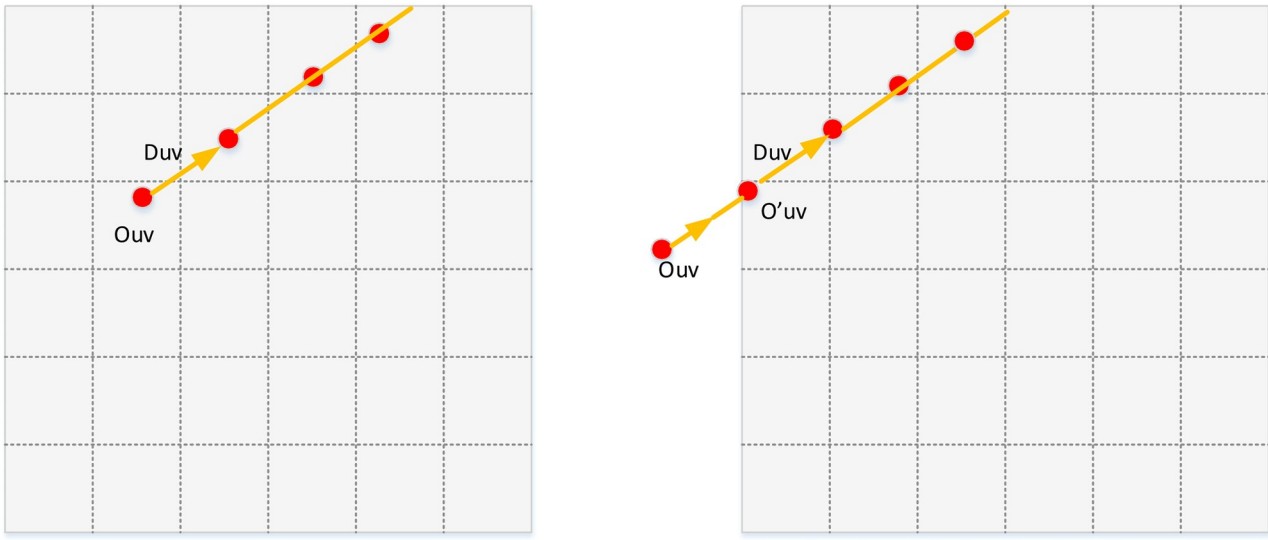

**Fig 5. Define view ray (which projected from the camera through the epipolar sample) by origin and orientation in shadow map space.**

minimum values as the thresholds. If its maximum value is is under the minimum depth saved at the 1D min-max tree, then the ray segments are completely lit. If the minimum of threshold is greater than the maximum depth saved at the 1D min-max tree, then the ray segments is shadowed. Otherwise, tthe step-by-step transform to the lower layer are performed when the segments are neither totally lit nor shadowed. Nodes that are completely in shadow are identified as dark blue, nodes that are completely in light are white, and nodes that contain visibility boundaries are gray, as shown in Fig 6c.

### Interactive participating media density estimating

Up to this point, we efficiently render the atmospheric scattering and volumetric light with minimal but sufficient samples in the low-level pipeline. To realize the interactions with participating media and multiple light sources, we need first to calculate the density of the participating media at every sample corresponding to a volumetric texture. There are various methods to generate air media, from Gaussian blur to physically driven method used in offline renderers. Also, there are various real-time solutions for more complex scattering functions [41]. This novel theoretical result enables the analytical computation of exact solutions to complex scattering phenomena which achieve an impressive interaction between light and participating media. However, these implementations are used for closed-form space rather than the outdoor open space and to compare with the proposed approach, their methods are far from real-time performance.

The proposed approach implies a LOD improvement by adjusting air density according to distance between the center of air particle and the view ray. For every samples, the ray is traced to generate noise, with the threshold $\varphi$ which serves as a random component for details of fog or cloud surface as shown in formula below:

$$\phi \leftarrow e^{\left(-\frac{\|pos-center\|}{r((1-ratio)+2*ratio*f(x,y,z))}\right)} \tag{1}$$

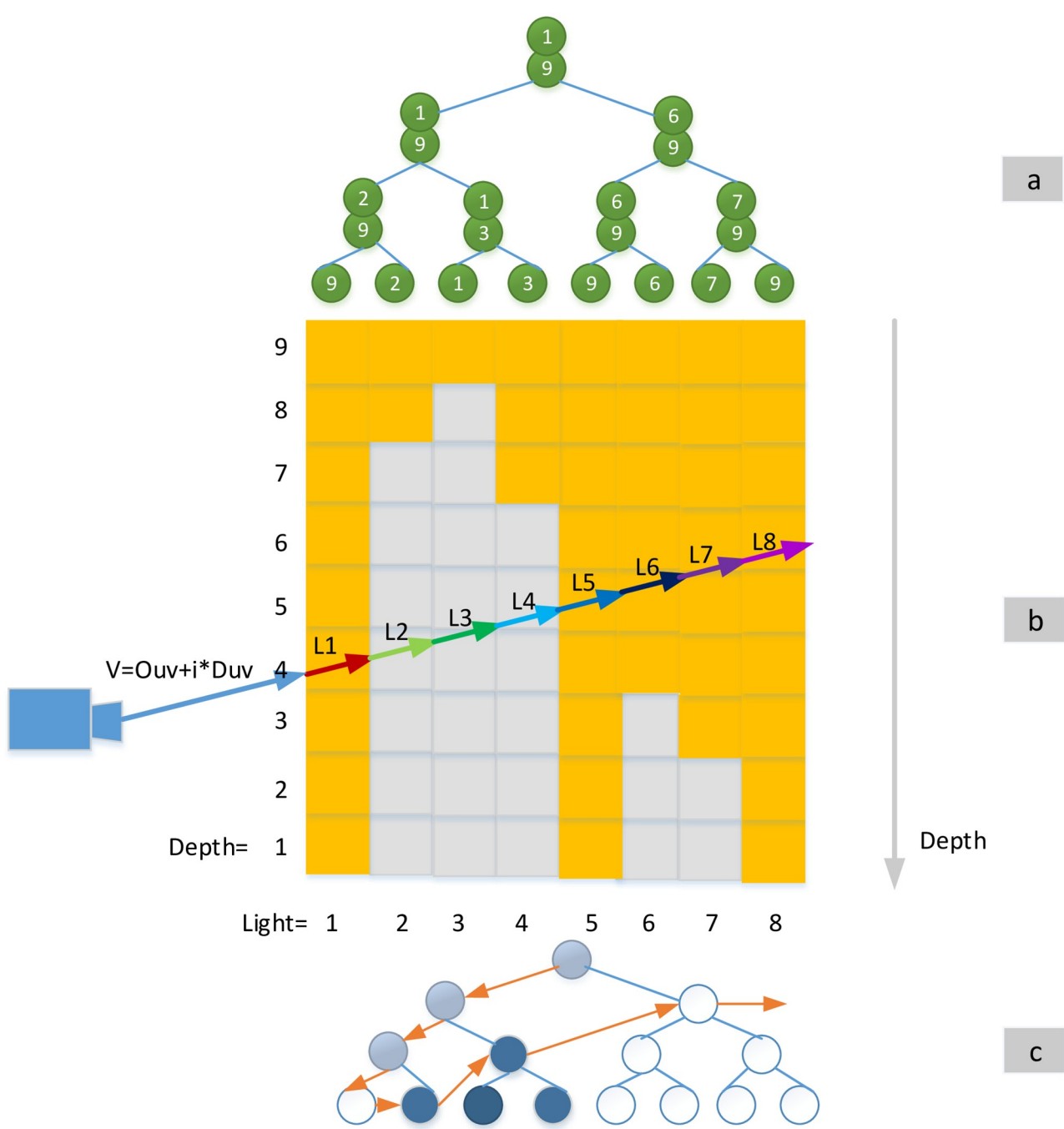

**Fig 6. The binary tree for detecting visible vector.**

The formula filters the unitized density and forms a sphere by zooming the radius ratio, where r is the radius of the sphere. Pos is the Euclidean straight line point of the ray. The f function is used to normalize the random variable with Perlin noise [39]

Further, 3D Gaussian distribution is used to calculate the position of pseudo-spheroids instead of particle system by Huang B et al. [40]. We use fewer primitives to generate more

details of cloud or fog boundary, as follows:

$$\begin{cases} Pos_x = M_x + \sim U(\alpha_x + \beta_x) \\[6pt] Pos_y = M_y + \sim U(\alpha_y + \beta_y) \\[6pt] Pos_z = M_z + \sim U(\alpha_z + \beta_z) \\[6pt] R_i = \frac{\varepsilon}{|Pos_x| + |Pos_y| + 1.0} \end{cases} \quad (2)$$

Where Pos is the position of the pseudo-spheroid. M is used as the center of the fog. U represents the random variable with normal distribution, $\alpha$ the mean value, Ri the distance from P to the C. $\varepsilon$ is applied to compute the magnitude of radius, and $\beta$ the standard deviation.

The interaction of multiple light sources significantly improves realism, but at the same time greatly increase the performance consumption. The numerical integration and the lookup table are designed to solve this issue. We observe that the optical depth relies on two variables: the height h and the angle $\varphi$ between gravity and the sun rays. So the Optical depth $T$ $(A \rightarrow P)$ can be pre-calculated and saved to the query form as the texture in GPU pipeline. Further, $\beta_R^e$ and $\beta_M^e$ can be represented as a 3D vector. Also the 2D texture is used to store the density of participating media from the ground to the top of the atmosphere. Finally, the optical depth integral is rewritten as follows:

$$T(A \rightarrow P) = \beta_R^e.xyz * T[h, \cos(\varphi)].x + \beta_M^e.xyz * T[h, \cos(\varphi)].y \quad (3)$$

However, the above optimization cannot support the additional cost by the calculation of multiple light sources. Further optimization is performed by 3D volumetric textures, which have 4-channels with a 16-bit floating-point type. By introducing a compute shader, the participating media density pass and the light scattering pass can be integrated together which avoids the overhead of frequently reading the memory. Moreover, all scattering is calculated locally, thus saving an immediate density buffer, which is beneficial for the situation of participating media density.

We march along the camera ray, accumulate the in-scattering coefficients and participating media density in parallel by 3D volumetric texture. Let us take the calculation of internal scattering coefficient as an example. The 3D texture is executed as an array of 2D textures, as shown in Fig 7.

## Results

### Data description

In this research, we gathered the Scene data from more than 40 civil airports in China, including Beijing and Shanghai. The raw dataset is also augmented by filtering and modifying to satisfy 3D modeling standards. Moreover, during data processing, the dataset is divided into three subsets three subsets: the elevation, texture, and 3D model datasets, accounting for 43%, 34%, and 23%, respectively. To test the stability of the proposed architecture, we reproduce a Flight track of two days from Chengdu airport (ZUUU) in China. We also simulate the time of day (TOD) to check the related algorithms in the proposed architecture.

### Qualitative tests

A set of tests is performed on the algorithm suite using nVidia GTX 980, running on a 64-bit i-Core 5 CPU 3.3GHz with 8 GB random access memory(RAM). The project is implemented

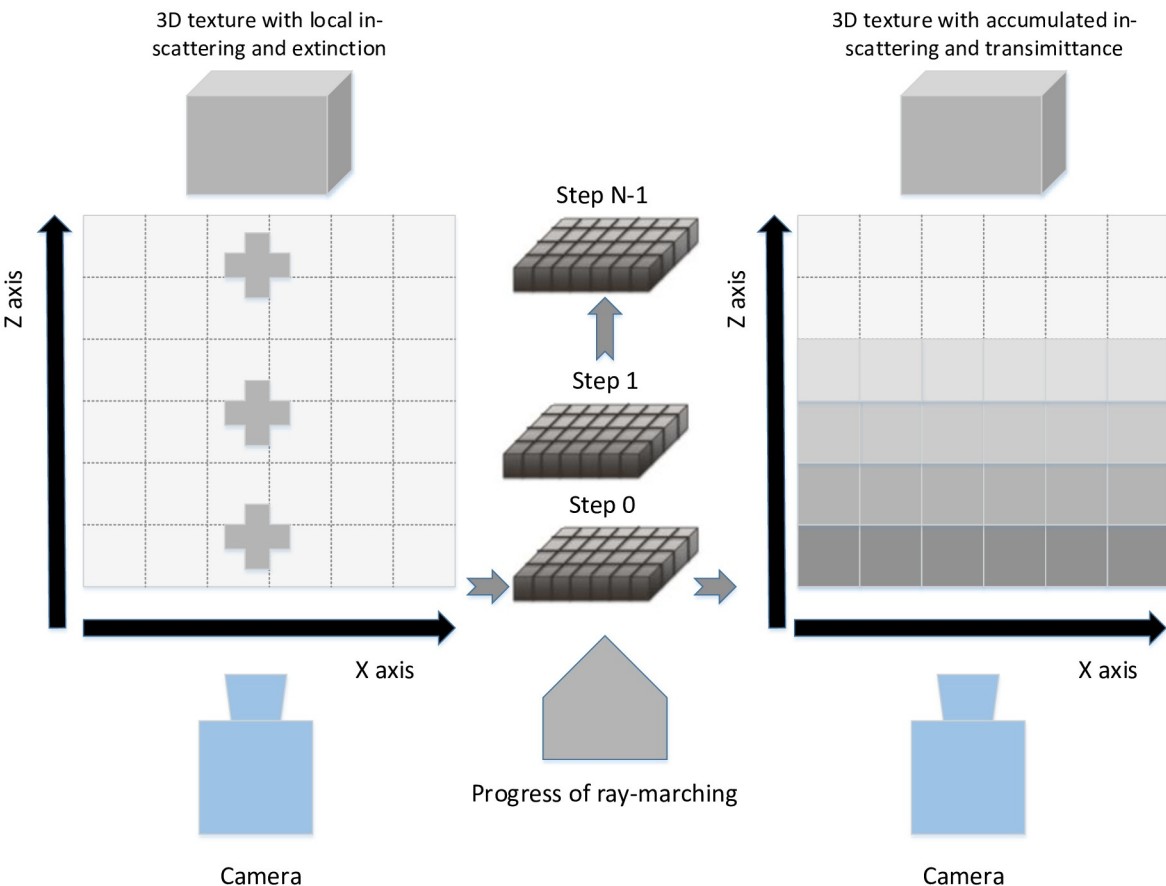

**Fig 7. 3D texture for accumulating scattering coefficient and paticipating media density.** a) At slice i (from 0 to N-1), read the coefficients of in-scattering and transmittance. b) Add the above coefficient and transmittance to the accumulating texture. c) Write out the accumulated in-scattering and transmittance to another volumetric texture at the same position. d) Increase i and proceed back to Step a.

entirely in C++ using the OpenGL for the host side and the OpenGL Shading Language (GLSL) for the GPU side. The ray-marching step size is determined by the adaptive Algorithm. The same algorithm suite performs perfectly in all resolutions using the nVidia GTX 980. The results of Qualitative experiments are illustrated in Figs 8–11 from which better realism, immersion and interactivity are observed. The realism of atmospheric scattering is improved by turning on the switch of proposed approach, as shown in Fig 8.

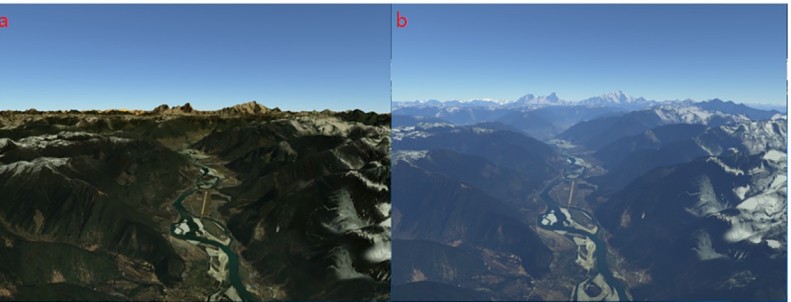

**Fig 8. Atmospheric scattering.** (a)Atmospheric scattering off. (b)Atmospheric scattering on.

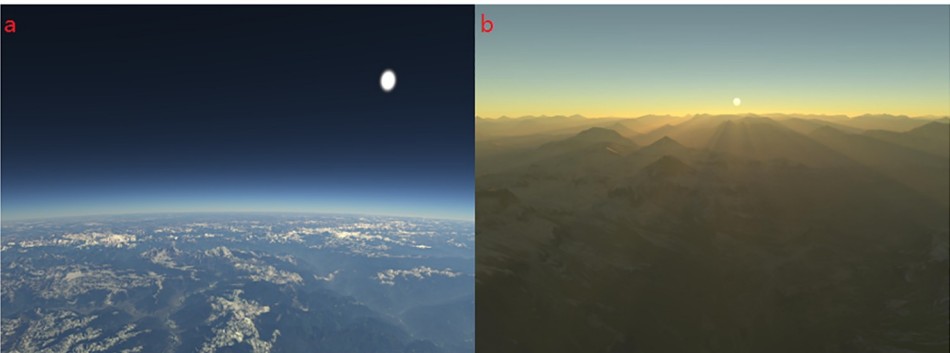

**Fig 9. Atmospheric scattering and volumetric light.** (a) Atmospheric scattering effect near the horizon. (b) Volumetric light and terrain shadow.

Then the test is performed when the camera is far away from the horizon where we can observe the boundary of the ellipsoid as shown in Fig 9a. Moreover, volumetric light interacts with mountains and terrain, which generate beams of light known as god rays, or crepuscular rays as shown in Fig 9b.

Then, the improvement of the atmospheric scattering at different time of the day is obtained by the proposed approach. The blue and purple light with shorter wavelengths is scattered away, which leads to orange or gold boundary in the evening, as shown in Fig 10.

The experiments are conducted on different scenarios with the equal time. For our proposal, we evaluate the performance obtained by anti-aliasing effect. From Fig 11, we can see

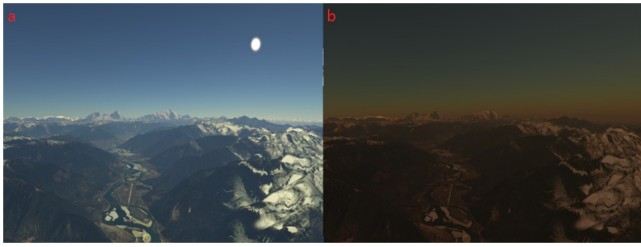

**Fig 10. Time of day.** (a)Atmospheric scattering effect during the day. (b)Atmospheric scattering effect in the evening.

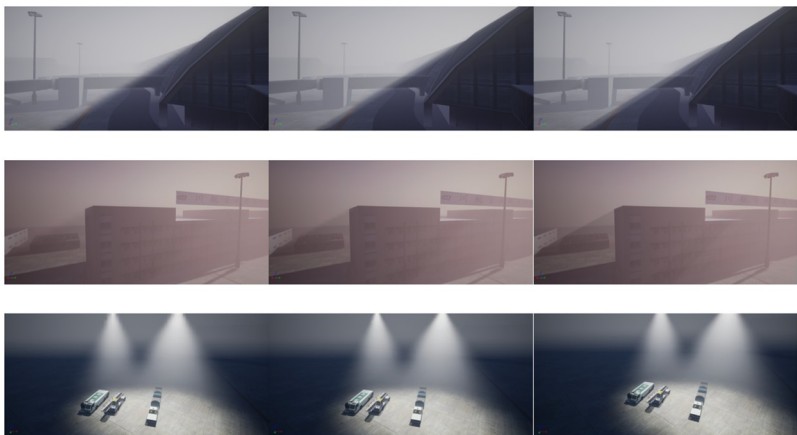

**Fig 11. A quality comparison with equal rendering time.**

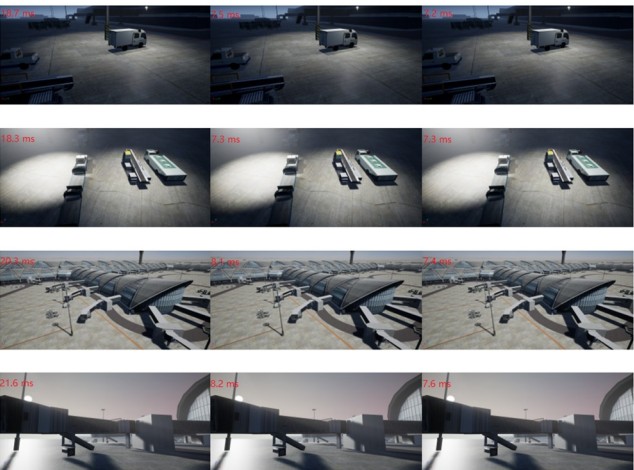

**Fig 12. A speed comparison with equal quality.**

that the experiments are divided into three parts for various scene and each row represents the results of the three methods: A (Chen et al. 2011), B (AliHH et al. 2016), and C (the proposed approach) from left to right. Regarding the different approaches, the proposed approach generates image with the least aliasing in the equal time.

The performance of the B (subsampling-based) is better than that of the A (1D mipmaps-based methods) because it is very good at improving performance by reducing samples, and making up for image quality by Bilateral Filtering. In contrast, although B achieve very good performance through subsampling, we realize better performance by adaptive sampling and a series of optimizations, while maintaining complete distribution of sampling points. The rendering time is shown in the upper left corner of the Fig 12. It can be seen from the experimental results that the best result is obtained by the proposed model with i.e., a 184% and 7.89% improvement against the A and B, respectively.

Fig 13 shows the density change of the participating media and the interaction with multiple light sources. Different from the above two experiments, A represents approach by Wyman et al. 2013 and B means method by Pegoraro et al. 2010. The results demonstrate the

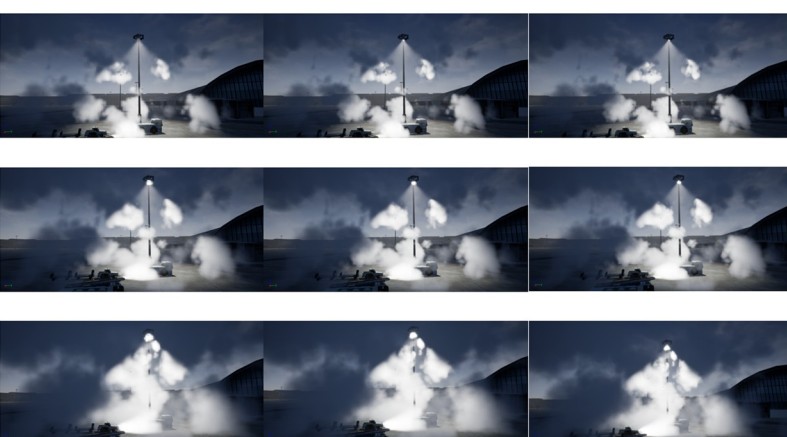

**Fig 13. An quality comparison by variable density of participating media with equal rendering time.**

significance the variable density in improving the performance of interactivity between the air media and volumetric light. We can observer that different effects and boundaries of the interaction between variable density and volumetric light, which leads to providing clearer distance hints and immersion.

## Quantitative tests

In this research, we apply the frames-per-second (FPS) and rendering time per frame (RTPF) to evaluate the performance of the adaptive volumetric light framework.

To evaluate the performance of atmospheric scattering and volumetric light framework, several experiments are conducted, the results of which are reported in Tables 1, 2 and 3. From the experimental results, we can draw the following conclusions:

a) The Brute forece method performs ray marching pixel-by-pixel, which is used as a benchmark for performance comparison. Comparatively, the ES approach by Engelhardt et al. achieves an improvement of 84.6 milliseconds accomplishing a single frame and the 1D min-max binary tree by Chen et al. further shortened 29.3 milliseconds, as shown in Table 1. And the Soft bilateral filtering shadows method by AliHH et al. obtains soft shadow with the lowest number of samples and the image quality is compensated by bilateral filtering. Compared with the proposed approach, the first and foremost, both methods achieve a good balance between efficiency and image quality. This is the reason that the method is very close to the proposed approach by 7.5 milliseconds. However, this method reduces the samples for efficiency, which leads to sparse distribution of ray marching, where some unnatural light beams will appear. The best result is obtained by proposed architecture within 7.2 milliseconds for rendering a complete frame by two more stages (Multi-light and Density).

b) 60 FPS is usually used as the FS application real-time performance benchmark. Although the 1D min-max binary tree by Chen et al. greatly improves the performance aginst the previous two methods, it still fails to achieve real-time performance by 38.8FPS in 1K resolution. By contrast, the Soft bilateral filtering shadows method by AliHH et al. and the proposed approach reach 60.6 and 67.6 FPS in 4K resolution, respectively.

c) Config1, Config2, and Config3 represent the various distribution of samples. The results demonstrate the significance of adaptive sampling in improving realism and interactivity, as shown in the picture above. Further, we describe the final image quality of various methods by quantifying the number of samples. From the experimental result, we can draw the following conclusions: Compare with the previous four methods, our proposed approach with confi1 improves the maximum number of samples by 336%, 255%, 166% and 45% at the same frame rate, respectively. Further, the maximum number of samples is improved by approximately 4.65% and 6.67% with config2 and config3, respectively. The results demonstrate the significant potential to achieve better image quality and efficiency with different configurations of sample distribution.

**Table 1. Performance of the methods at various rendering stage in a single frame.**

| Methods | Shadowmap | OpticalDepth | Binarytree | Multi-light | Density | Integration | Total |
|---|---|---|---|---|---|---|---|
| *Brute force* | 73.2 | 23.2 | N/A | N/A | N/A | 36.3 | 132.7 |
| *Engelhardt T et al.* 2010 | 19.2 | 13.1 | N/A | N/A | N/A | 15.8 | 48.1 |
| *Chen et al.* 2011 | 6.2 | 4.2 | 1.7 | N/A | N/A | 6.7 | 18.8 |
| *AliHH et al.* 2016 | 3.2 | 1.8 | 1.2 | N/A | N/A | 1.3 | 7.5 |
| *the Proposed method* | 3 | 1.6 | 0.9 | 0.4 | 0.4 | 0.9 | 7.2 |

**Table 2. FPS of methods with different resolutions.**

| Methods | Resolution 1K | Resolution 2K | Resolution 4K |
|---|---|---|---|
| *Brute force* | 7.5 | 4.8 | 2.9 |
| *Engelhardt T et al.* 2010 | 20.8 | 12.8 | 7.3 |
| *Chen et al.* 2011 | 53.2 | 21.6 | 14.5 |
| *AliHH et al.* 2016 | 133.3 | 88.5 | 60.6 |
| *Theproposedmethod* | 138.9 | 95.2 | 67.6 |

**Table 3. Numbers of samples supported by each method ah the same FPS.**

| Methods | Samples at levle 1 | Samples at levle 2 | Total Samples |
|---|---|---|---|
| *Brute force* | 3 | 19 | 22 |
| *Engelhardt T et al.* 2010 | 4 | 23 | 27 |
| *Chen et al.* 2011 | 6 | 32 | 36 |
| *AliHH et al.* 2016 | 12 | 54 | 66 |
| *Config*1 | 43 | 43 | 86 |
| *Config*2 | 30 | 60 | 90 |
| *Config*3 | 26 | 70 | 96 |

## Discussion

From the experimental results, we can see that the proposed framework obtains a better sense of reality in real-time. At the same time, our adaptive sampling-based methods can be well adapted to applications where the complexity and scope of the scene change dramatically.

To improve the performance, some assumptions are made about the scattering. Only a single scattering is taken in to account for achieving an interactive rate. For smoothly varying media, only a small number of samples are calculated and interpolated within the transparent or translucent media. We also tested the proposed methods with more light sources. When the number of light sources exceeds 10, the performance drop sharply, especially when translucent media is taken into account. Like all algorithms based on shadow map, the performance is also limited by its resolution. Although adaptive sampling reduces the above impact to a certain extent, it is still the optimization for future work. Moreover, the proposed method realizes the real-time interaction between volumetric light and a semi-transparent media, but the self-shadow of semi-transparent medium is not completed.

## Conclusion

In this work, to achieve realism and immersion by volumetric light and atmospheric scattering in real-time, we present a dynamic range of volumetric light architecture by introducing adaptive sampling-based and series optimization. According to the distance from the camera, we design different pipelines with targeted sampling step size and sampling strategies. To the best of our knowledge, this is the original study to introduce dynamic range and adaptive sampling-based architecture for volumetric light using IPMDE approaches. The sampling step size and distribution are taken into account by mearing distance from camera. The tasks in VFC are accomplished by the faster 1D binary tree, which prevents shadowed samples from participating in expensive calculating. In the IPMDE method, the interaction between participating media and multiple light sources is realized by 3D texture and the lookup table. The

experimental results in this approach demonstrate that the proposed framework can serve as a significant application for current FS.

## Future work

In our future work, we plan to consider the multiple scattering instead of single scattering in our architecture. More advanced optimizations are expected to propose. The shadow of the translucent is also a problem that needs to be addressed in this framework.

## Supporting information

**S1 Appendix. Basic priciples.**
(PDF)

**S1 Video. Demostration video.**
(MP4)

**S1 Data.**
(TXT)

**S1 Dataset.**
(RAR)

**S1 Sourcecode.**
(RAR)

## Author Contributions

**Conceptualization:** Zhang jianwei.

**Investigation:** Lin yi.

**Methodology:** Ge wenyi.

**Project administration:** Liu hong.

**Resources:** Yang menglong.

**Software:** Tan shihan.

**Writing – original draft:** Tan shihan.

**Writing – review & editing:** Tan shihan.

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
