## [Decision Letter · Decision Letter 0]

21 Aug 2020

PONE-D-20-19144

Adaptive volumetric light and atmospheric scattering

PLOS ONE

Dear Dr. ShiHan,

Thank you for submitting your manuscript to PLOS ONE. After careful consideration, we feel that it has merit but does not fully meet PLOS ONE’s publication criteria as it currently stands. Therefore, we invite you to submit a revised version of the manuscript that addresses the points raised during the review process.

Based on the comments received by reviewers, the editor’s decision is “major revision” primarily due to the following reasons:

 More experimentation in different scenarios is needed to demonstrate the effectiveness of proposed method.Some closely-related papers are not cited.Lack of clarity in devising problem statement for proposed research from existing literature.

Please revise the paper by incorporating all reviewer’s comments.

We look forward to receiving your revised manuscript.

Kind regards,

Gulistan Raja

Academic Editor

PLOS ONE

Journal Requirements:

2. Please ensure that you refer to Figure 7 in your text as, if accepted, production will need this reference to link the reader to the figure.

Reviewers' comments:

Reviewer's Responses to Questions

**Comments to the Author**

1. Is the manuscript technically sound, and do the data support the conclusions?

Reviewer #1: Yes

Reviewer #2: Yes

Reviewer #3: No

2. Has the statistical analysis been performed appropriately and rigorously? 

Reviewer #1: Yes

Reviewer #2: N/A

Reviewer #3: Yes

3. Have the authors made all data underlying the findings in their manuscript fully available?

Reviewer #1: Yes

Reviewer #2: No

Reviewer #3: Yes

4. Is the manuscript presented in an intelligible fashion and written in standard English?

Reviewer #1: Yes

Reviewer #2: Yes

Reviewer #3: No

5. Review Comments to the Author

Reviewer #1: In this manuscript, the authors present a methodology for improving the visual representation (graphics) of a flight simulator, by simulating volumetric light and atmospheric scattering in a realistic and computationally efficient way. The methodology seems reasonable, and the results look very good. I have suggestions for some minor revisions, as follows.

1. The section "Related works" should be incorporated into the Introduction, and the Introduction should then end by making clear the distinction between what was accomplished in the related works and the specific methodology of the current work.

2. Points 7 and 9 on p. 4 are a bit vague, although the meaning becomes clearer once the reader gets to the detailed description of the algorithm.

3. Step b in the subsection "Epipolar sampling" on p. 4 is also vague. Specifically, what constitutes a sample, and what does "are marched" mean? Are the samples values at grid points where calculations are performed? Does their spacing determine the spacial resolution, or are calculations also performed as the samples are propagated between the grid points shown in the figures?

4. p. 5: "its direction D_uv" - It seems that D_uv should be a vector with dimensions of length and not simply a direction. Is that correct?

5. Fig. 6c is not entirely clear, specifically the arrows indicating transitions from lower levels to higher levels, which are not explained in the paragraph that references Fig. 6c in the text.

6. In Fig. 8, are (a) and (b) switched (or are their descriptions switched in the text)?

7. In Table 1, I presume the slashes indicate categories that are not applicable. Perhaps "N/A" would be clearer than just a slash.

8. The section "Discussion" seems to be too short as a section unto itself.

9. Overall, the writing is decent, but there are some English mistakes here and there. After editing for content, the manuscript should be proofread again to catch such English mistakes. Some examples are:

p. 1 "slice of disadvantages"

p. 2 "the sampling strategy and steps need to redesign"

p. 2 "And the reasonable..."

p. 2 "And then..."

p. 3 "While further..."

p. 4 "Near clipping plan" and "Far Clipping plan"

p. 6 "visible vator"

p. 6 "by introduced"

p. 6 "density of participating media density"

p. 7 "is taxonomically into three subsets"

p. 7 "two-days flight"

p. 7 "results of Qualitative experiments"

p. 7, "Besides, ..."

p. 9 "mearing distance"

in general, the use of capitalization in the figures

Reviewer #2: Q1: This manuscript describes a real-time software system for rendering atmospheric effects in large-scale outdoor scenes, in the context of a flight simulator. It builds on the existing techniques of Epipolar Sampling. The paper's results appear quite reasonable and look very nice, and the strategies being described seem to be valid approaches to solving the problem. The results are described reasonably and report the performance (timing) measurements that are needed.

However, I would describe the description of the system as rather cursory and incomplete. It would not be possible to reproduce the results from the level of detail that appears in the paper. The precision and completeness with which the method is explained and derived is markedly lower than other work in the area, e.g. refs 12 and 19 and the following paper that is not cited:

J. Chen et al. "Real-time volumetric shadows using 1D min-max mipmaps," I3D 2011, doi:10.1145/1944745.1944752

The results of the system are not compared to any ground-truth references (e.g. by slow path tracing methods) to show how accurate they are, nor is the accuracy compared to previous methods (only the performance, and only to reference 12).

Because of this it is difficult to authoritatively judge correctness. Though it must be said that in the real time context, correctness in practice often means that it produces good-looking images fast enough, which this system does seem to.

Q3: As I read the expectations of PLOS ONE for data availability, I think the appropriate interpretation for a graphics paper is that the authors release an open source implementation of the method, together with the models required to replicate the key results of the paper. The authors simply state that all required data is in the paper, but I don't think this meets the spirit of making the data available that is needed to evaluate the paper's claims about its method.

Q4: The English is not very standard, but I did not have any trouble understanding it, so I would not think this should be a barrier for this manuscript.

On the whole, though it's not obvious to me exactly how PLOS's criteria apply to a computer graphics paper, I think it is safe to say that the paper does not meet the expectations for completeness, validation, and comparison against prior art that would be expected by a journal or peer-reviewed conference in computer graphics. The paper does not clearly state what technology is new and what is adopted from prior art, and it provides no comparisons to illustrate the benefits of the new components introduced in this manuscript. Also, it fails to cite the paper noted above (though it does cite a related one) which seems to overlap closely with the min/max tree methods used here for adaptive integration along epipolar lines.

I have made the recommendation "major revision" because I do believe a publishable paper can be written about this system. It needs to include a complete and clear description of the system and the algorithms it uses, clearly discuss which of those algorithms are new, demonstrate the benefits provided by the new methods, validate the accuracy of the images being produced, and (if my interpretation of the data policy is correct) include code and data that can be used to replicate the results.

Reviewer #3: Volumetric lighting is a hard problem in real-time rendering. This submission presents a technique to add it to a flight simulator, which has more rigid performance constraints than many real-time applications. This paper presents the engineering decisions made for a particular implementation, which is a reasonable publishable contribution.

However, there are some problems with the submitted work.

The first major problem is is unclear how the authors want it evaluated. The text suggests it is a novel research algorithm that solves what the authors claim were previously unachievable effects. The video and images suggest it should be evaluated as a simulation that faithfully and accurately reproduces the images see from an airplane cockpit, without worrying about specific complex scattering and shadowing. As a reviewer, I read these differently. The first is a research paper, the second is an engineering / systems paper. Unfortunately because of this lack of clairty, in the current state, I couldn't recommend it for acceptance in either category.

The second major problem is the paper skips citing many closely-related papers. For instance, the work on "Voxelized Shadow Volumes" by Chris Wyman (from High Performance Graphics 2011) and the follow up "Imperfect Voxelized Shadow Volumes" from HPG 2013 use a very similar epipolar space sampling technique that aligns ray marching samples with the alignment of samples in memory, and this scales to many lights (at least in the "imperfect" case). While other work by Chen et al. is cited, the paper "Real-Time Volumetric Shadows using 1D Min-Max Mipmaps" from Symp. on Interactive 3D Graphics 2011 is not cited, and their collective work renders volumetric shadows on rectified shadow maps using 1D mipmaps / binary trees (similar to the approach described in the text). Citing the Tevs et al I3D 2008 paper "Maximum mipmaps for fast, accurate, and scalable dynamic height field rendering" would also be appropriate given the use of 1D trees for traversal. Also, it's not clear why a Gaussian blob model was chosen for heterogeneous media. There are a variety of models that can be used for media ranging from Gaussian radial basis functions up to complex simulation-driven formats used in film renderers. I would have hoped to see some discussion for the rationale and citations of the relevant related work. Certainly, the real-time work typically relies on homogeneous media, but there is more exploration of more complex alternatives in the offline rendering literature. And there are various real-time solutions for more complex scattering functions (e.g., Pegararo et al's work "A closed-form solution to single scattering for general phase functions and light distributions" from Computer Graphics Forum 29(4).)

The third problem with the paper is the fairly simplistic evaluation. One of the key features of this work, as far as I can understand, is the handling of shadowing and visibility inside the media. The only image in the paper that clearly shows shadows is Figure 9b. And this is a scene where a radial image-space filter emanating from the sun would give nice results (this is frequently what game developers used for volumetric shadows today). It is important to see how the proposed approach from the paper works in a variety of scenarios and not just fairly simple situations. Additionally, one key feature of the paper is that it applies to not just one light source, but to many light sources. Only one example is provided in the submission (in Figure 11), and as far as I can tell this image does not have any shadows. Essentially: the results are unclear if this algorithm works for providing shadows from many lights. (Or if "shadows" and "many light source" are two independent improvements that do not work together.) Additionally, as far as I can tell, the only change in media density is the fairly simplistic decrease in density with altitude (shown in Figure 9a). It is unclear if the proposed approach in Equations 1, 2, and 3 is evaluated. Certainly, the texture ping-pong / ray-marching approach described in lines 219-223 is probably unnecessary with media density that changes so smoothly. The video isn't particularly useful -- it shows only simplistic relatively homogeneous media, as far as I can tell, without any shadowing, and without any ground truth comparisons. Ideally, for a flight simulator, you could compare with actual photos in similar situations.

A fourth problem with this submission is it skips the most important aspect of a research paper (in my opinion): describing why the work is needed. Typically, one would use a known, existing algorithm unless it does not solve a problem required by your application (i.e., your flight simulator). There is no need to develop a new algorithm that may have new and unknown problems without a good reason. These reasons should be clearly specified, and the choices made by a novel algorithm should derive from the functionality desired. This paper does not tie together the algorithm with the rationale, it simply specifies the algorithmic steps in a bulleted list without describing why these choices are better than any possible alternatives. Once you motivate each step in the new algorithm, it is much easier to judge whether the proposed approach will significantly improve over prior art. (Also, it often makes it easier to design evaluation images that clearly show the improvement.)

As far as clarity of the text, I would provide the following suggestions:

1) Provide an overview section that summarizes the new approach and why this new approach addresses the failures of prior work

2) Revisit the enumerated contributions (currently on lines 47-60). Make sure each one is measurable. Make sure theses measurements are provided in the evaluation.

3) For each algorithmic section, provide some reminder of what the next step in the algorithm needs to solve, and discuss which aspects are new and which reuse ideas from prior work.

6. PLOS authors have the option to publish the peer review history of their article (what does this mean?). If published, this will include your full peer review and any attached files.

Reviewer #1: **Yes: **Carynelisa Haspel

Reviewer #2: No

Reviewer #3: No

---

## [Author Response · Author response to Decision Letter 0]

27 Sep 2020

This submission contains one-by-one responses to the review comments, see "Rebuttal letter" for details

---

## [Decision Letter · Decision Letter 1]

6 Oct 2020

PONE-D-20-19144R1

Adaptive volumetric light and atmospheric scattering

PLOS ONE

Dear Dr. ShiHan,

Thank you for submitting your manuscript to PLOS ONE. After careful consideration, we feel that it has merit but does not fully meet PLOS ONE’s publication criteria as it currently stands. Therefore, we invite you to submit a revised version of the manuscript that addresses the points raised during the review process.

The reviewer 2 found that all comments have been addressed which were raised by him in last cycle of review. He was of the view that manuscript is near the right condition for minor-revisions accept subject to fulfillment of some observation raised by him in his review recommending major revisions. On the other hand, Reviewer 1 is satisfied with the revised version of manuscript and recommended minor revisions.

After thorough consideration of comments of both reviewers, my decision is minor revisions. Please incorporate all the comments raised by both reviewers.

We look forward to receiving your revised manuscript.

Kind regards,

Gulistan Raja

Academic Editor

PLOS ONE

Reviewers' comments:

Reviewer's Responses to Questions

**Comments to the Author**

1. If the authors have adequately addressed your comments raised in a previous round of review and you feel that this manuscript is now acceptable for publication, you may indicate that here to bypass the “Comments to the Author” section, enter your conflict of interest statement in the “Confidential to Editor” section, and submit your "Accept" recommendation.

Reviewer #1: (No Response)

Reviewer #2: All comments have been addressed

2. Is the manuscript technically sound, and do the data support the conclusions?

Reviewer #1: Yes

Reviewer #2: Yes

3. Has the statistical analysis been performed appropriately and rigorously? 

Reviewer #1: Yes

Reviewer #2: N/A

4. Have the authors made all data underlying the findings in their manuscript fully available?

Reviewer #1: Yes

Reviewer #2: Yes

5. Is the manuscript presented in an intelligible fashion and written in standard English?

Reviewer #1: No

Reviewer #2: No

6. Review Comments to the Author

Reviewer #1: This is a revision of manuscript PONE-D-20-19144. From the reviews of manuscript PONE-D-20-19144, the reviewers, including myself, made suggestions that were based mostly on properly framing the work that the authors’ have completed in comparison to previous work, providing more and clearer detail regarding the authors’ algorithm (including providing the source code itself), and providing more demonstrations of the success of the authors’ algorithm. The authors have made numerous changes to the manuscript in response to those suggestions. In the revised manuscript, the reader now has a better perspective on the novelty of the authors’ work in comparison to previous work, and some important references to previous work that were missing in the original submission have now been included. The subsections titled “Related works” and “Overview” are a little unconventional, and the Introduction could have been structured differently while including the same information, but the necessary content is now there. The description of the method is now more detailed and clearer. The results from the previous submission looked good, and the additional results that the authors have included in the current submission, including comparisons to ground-truth references, also look good.

I think the paper could still use some minor revisions in terms of editing for English, especially in the newly added sections. Just a couple of examples include:

(1) The use of the word “finally” is incorrect in a number of places in the text.

(2) There are mistakes in capitalization, such as “chen” on line 277, “Transversed” on line 285, and “Textures” on line 357.

Reviewer #2: This revision is commendably responsive to the reviews. The manuscript is much improved in terms of detailed description of the contributions and proposed methods, and in evaluation of the results of those methods. Good discussion of the contributions of previous work relative to the current paper is included, and detailed pseudocode is provided to clarify the details for those who wish to implement the method. For me the level of novelty and the improvements in performance shown here are sufficient for a publication, and subject to the caveats below the paper appears to be near the right condition for a minor-revisions accept. But some of the below involve enough uncertainty that I'm still at "major revision".

Some things still missing:

* The source of the implementations of Chen et al. and Ali et al. is not mentioned. It should be clarified that it's the authors' own reimplementation (if it is), any differences with exactly what was proposed in those papers should be described, and these reimplementations should be included in the released source code.

* The source code contains comments in Chinese, many of which seem to be translations of the English but some of which seem to include new information; for publication everything should be in English.

* The text mentions a compute shader which I do not see in the source code.

* The understandability of the English has become worse in this revision, and I think it now needs some work to bring it up to the standard of "intelligible" everywhere. Really the paper should get a thorough edit by someone with more experience in written English.

It's worth pointing out that the source code does not meet the standard of replicability, since it cannot be compiled without the rest of the system from which it comes. But to the extent it contains the full implementation of the proposed method, I think it does meet the standard of revealing exactly what was used to compute the results. Including the implementations of the competing methods would make this more complete.

Some things that have me worried:

* The timing results appear to be from the same experiments as before, but with the methods relabeled. It's good that they are now labeled in a way that credits the pieces to earlier papers, but I am a little concerned that the Chen et al and Ali et al papers, which were not previously referenced, are now the labels for results that were included before. Were these methods implemented before but not cited? Or is the implementation that was made without referencing these papers now being used to represent those methods in the comparison? Both interpretations are a problematic.

* The paper by Ali et al. (formerly just called "subsampling" without any explanation) is included in the comparison but not discussed at all in the prior work. It also seems to produce the closest results to the new method. The differences between and relative merits of the methods need to be discussed.

* The authors have borrowed two complete sentences of text from the reviews and incorporated them into the discussion of prior work: "...a very similar epipolar space sampling technique that aligns ray marching samples with the alignment of samples in memory, and this scales to many lights..." and "There are a variety of models that can be used for media ranging from Gaussian radial basis functions up to complex simulation-driven formats used in film renderers...". This text is not the authors' work; reviewers are not here to write papers for authors. I'm not sure what PLOS's policies say about this.

7. PLOS authors have the option to publish the peer review history of their article (what does this mean?). If published, this will include your full peer review and any attached files.

Reviewer #1: **Yes: **Carynelisa Haspel

Reviewer #2: No

---

## [Author Response · Author response to Decision Letter 1]

28 Oct 2020

Cover Letter 

Dear Editor and reviewers 

Thank you very much for giving us an opportunity to revise our manuscript. We 

appreciate the editor and reviewers very much for their constructive comments and 

suggestions on our manuscript entitled “Adaptive volumetric light and atmospheric 

scattering” (ID: PONE-D-20-19144R1). 

 We have studied reviewers’ comments carefully. According to the reviewers’ 

detailed suggestions, we have made a careful revision on the original manuscript. All 

revised portions are marked in highlight in the revised manuscript which we would like 

to submit for your kind consideration. 

Kind regards. 

TanShihan 

E-mail: 530916232@qq.com

Rebuttal Letter 

Dear Editor and reviewers: 

Thank you for your letter and the reviewers’ comments on our manuscript entitled 

“Adaptive volumetric light and atmospheric scattering” (ID: PONE-D-20-19144R1). 

Those comments are very helpful for revising and improving our paper, as well as 

the important guiding significance to other research. We have studied the comments 

carefully and made corrections which we hope meet with approval. The main 

corrections are in the manuscript and the responds to the reviewers’ comments are as 

follows (the replies are highlighted in blue). 

Replies to the reviewers’ comments: 

Reviewer #1: 

Q1. This is a revision of manuscript PONE-D-20-19144. From the reviews of manuscript 

PONE-D-20-19144, the reviewers, including myself, made suggestions that were based 

mostly on properly framing the work that the authors’ have completed in comparison to 

previous work, providing more and clearer detail regarding the authors’ algorithm (including 

providing the source code itself), and providing more demonstrations of the success of the 

authors’ algorithm. The authors have made numerous changes to the manuscript in response 

to those suggestions. In the revised manuscript, the reader now has a better perspective on 

the novelty of the authors’ work in comparison to previous work, and some important 

references to previous work that were missing in the original submission have now been 

included. The subsections titled “Related works” and “Overview” are a little unconventional, 

and the Introduction could have been structured differently while including the same 

information, but the necessary content is now there. The description of the method is now 

more detailed and clearer. The results from the previous submission looked good, and the 

additional results that the authors have included in the current submission, including 

comparisons to ground-truth references, also look good. 

I think the paper could still use some minor revisions in terms of editing for English, 

especially in the newly added sections. Just a couple of examples include: 

(1) The use of the word “finally” is incorrect in a number of places in the text. 

(2) There are mistakes in capitalization, such as “chen” on line 277, “Transversed” on line 

285, and “Textures” on line 357. 

Response：(1) We have revised the use of conjunctions, including but not limited to 

“finally”, see the highlighted part for details. 

(2) We have modified these mistakes in capitalization, such as “chen” ->”Chen” on line 

246, “Traversed”->” traversed” on 255 and “Textures”->”texture” on line 333. 

Reviewer #2: 

Q1. This revision is commendably responsive to the reviews. The manuscript is much 

improved in terms of detailed description of the contributions and proposed methods, and 

in evaluation of the results of those methods. Good discussion of the contributions of 

previous work relative to the current paper is included, and detailed pseudocode is provided 

to clarify the details for those who wish to implement the method. For me the level of novelty 

and the improvements in performance shown here are sufficient for a publication, and 

subject to the caveats below the paper appears to be near the right condition for a minor-

revisions accept. But some of the below involve enough uncertainty that I'm still at "major 

revision". 

* The source of the implementations of Chen et al. and Ali et al. is not mentioned. It should 

be clarified that it's the authors' own reimplementation (if it is), any differences with exactly 

what was proposed in those papers should be described, and these reimplementations 

should be included in the released source code. 

Response： Unfortunately, the source code of Chen et al. is not found. We re-

implement it on basis of other 1D min/max mipmaps source code on “gethub”, and 

modified it step by step according to the original paper. For the approach by Ali et al., 

we find its implement. The source code of the above two articles have been added to 

this submission. 

Q2. * The source code contains comments in Chinese, many of which seem to be 

translations of the English but some of which seem to include new information; for 

publication everything should be in English. 

Response：Yes, the Chinese description in the source code has been changed to English 

for publication, see the new version source code for details. 

Q3. * The text mentions a compute shader which I do not see in the source code. 

Response：Because in the historical version, some components do not support the 

compute shader, we put all this type files in a separate folder, resulting in the missing 

of the compute shader part in the last version. This submission will include this one. 

Q4. * The understandability of the English has become worse in this revision, and I think it 

now needs some work to bring it up to the standard of "intelligible" everywhere. Really the 

paper should get a thorough edit by someone with more experience in written English. 

Response： Yes, an American student in this field was invited to standardize and 

improve the language description, see the highlighted part for details. 

Q5. It's worth pointing out that the source code does not meet the standard of replicability, 

since it cannot be compiled without the rest of the system from which it comes. But to the 

extent it contains the full implementation of the proposed method, I think it does meet the 

standard of revealing exactly what was used to compute the results. Including the 

implementations of the competing methods would make this more complete. 

Response： Yes, we add the implementations of the competing methods in the source-

code file, such as soft bilateral filtering shadows by Ali HH et al. and 1D min-max 

mipmaps approach by Chen et al. 

Q6. * The timing results appear to be from the same experiments as before, but with the 

methods relabeled. It's good that they are now labeled in a way that credits the pieces to 

earlier papers, but I am a little concerned that the Chen et al and Ali et al papers, which 

were not previously referenced, are now the labels for results that were included before. 

Were these methods implemented before but not cited? Or is the implementation that was 

made without referencing these papers now being used to represent those methods in the 

comparison? Both interpretations are a problematic. 

Response：I’m sorry that the previous description is not clear enough. In the previous 

version of the manuscript, the experimental comparison was divided into several 

categories according to the methods, without specifying which paper was used as the 

reference. Indeed, for the “ES+BT” type, the paper we actually referred to at the time 

was “Chen J, Baran I, Durand F, Jarosz W. Rendering images with volumetric shadows 

using rectified height maps for independence in processing camera rays; 2016.” 

(Number 28 in the references in our manuscript) This paper is the subsequent version 

of “Real-time volumetric shadows using 1D min-max mipmaps” by the same author in 

which ES and 1D min-max acceleration data structure for rectified shadow map is also 

the core algorithm. Unfortunately, the source code of these two paper is not found. 

We re-implement it on basis of other 1D min/max mipmaps source code on gethub, 

and modified it step by step according to the original paper. Since the implementation 

of the two versions of the experimental reference paper is essentially the same version 

we implemented, only the name has been modified, and the results of the experiment 

remain unchanged. Similarly, the approach by Ali et al. was called “ES+SS” in the 

previous version of manuscript, which is actually the same work by Ali et al in this 

version. Therefore, the experimental results have not changed, and the source code 

will be submitted in this version. 

Q7. * The paper by Ali et al. (formerly just called "subsampling" without any explanation) 

is included in the comparison but not discussed at all in the prior work. It also seems to 

produce the closest results to the new method. The differences between and relative merits 

of the methods need to be discussed. 

Response：Yes we discussed the paper by Ali et al. as follows: This study introduces 

the soft bilateral filtering shadows method of image-based shadows. Its main 

contribution is to obtain soft shadow with the lowest number of samples and the 

image quality is compensated by bilateral filtering. Compared with the proposed 

approach, the first and foremost, both methods achieve a good balance between 

efficiency and image quality. This the why the method is very close to the proposed 

approach. However, this method reduces the samples for efficiency, which leads to 

sparse distribution of ray marching, where some unnatural light beams will appear. 

See the highlighted part for details on line 408-414. 

Q8. * The authors have borrowed two complete sentences of text from the reviews and 

incorporated them into the discussion of prior work: "...a very similar epipolar space 

sampling technique that aligns ray marching samples with the alignment of samples in 

memory, and this scales to many lights..." and "There are a variety of models that can be 

used for media ranging from Gaussian radial basis functions up to complex simulation-driven 

formats used in film renderers...". This text is not the authors' work; reviewers are not here 

to write papers for authors. I'm not sure what PLOS's policies say about this. 

Response：Yes, we have re-modified the related description. 

"...a very similar epipolar space sampling technique that aligns ray marching samples 

with the alignment of samples in memory, and this scales to many lights..." 

to “… a very similar approach by using epipolar space sampling, which aligns the 

samples of ray marching in memory. Then, many lights are supported by their 

subsequent works” on line 135-137. 

"There are a variety of models that can be used for media ranging from Gaussian radial 

basis functions up to complex simulation-driven formats used in film renderers..." 

to” There are various methods to generate air media, from Gaussian blur to 

physically driven method used in offline renderers” on line 302-303. 

End of Reply--------------------------------------------- 

Once again, thank you very much for your constructive comments and suggestions 

which would help us both in English and in depth to improve the quality of the paper. 

Kind regards, 

TanShihan 

E-mail: 530916232@qq.com

---

## [Editor Report · Decision Letter 2]

30 Oct 2020

Adaptive volumetric light and atmospheric scattering

PONE-D-20-19144R2

Dear Dr. ShiHan,

We’re pleased to inform you that your manuscript has been judged scientifically suitable for publication and will be formally accepted for publication once it meets all outstanding technical requirements.

Kind regards,

Gulistan Raja

Academic Editor

PLOS ONE
---

## [Editor Report · Acceptance letter]

6 Nov 2020

PONE-D-20-19144R2 

Adaptive volumetric light and atmospheric scattering  

Dear Dr. ShiHan:

I'm pleased to inform you that your manuscript has been deemed suitable for publication in PLOS ONE. Congratulations! Your manuscript is now with our production department. 

Kind regards, 

on behalf of

Dr. Gulistan Raja 

Academic Editor

PLOS ONE